# A Search Engine Concept to Improve Food Traceability and Transparency: Preliminary Results

**DOI:** 10.3390/foods11070989

**Published:** 2022-03-29

**Authors:** Caterina Palocci, Karl Presser, Agnieszka Kabza, Emilia Pucci, Claudia Zoani

**Affiliations:** 1Department of Enterprise Engineering, University of Rome Tor Vergata, 00133 Rome, Italy; 2Premotec GmbH, 8400 Winterthur, Switzerland; karl.presser@premotec.ch (K.P.); agnieszka.kabza@premotec.ch (A.K.); 3Italian National Agency for New Technologies, Energy and Sustainable Economic Development (ENEA), Department for Sustainability, Biotechnology and Agroindustry Division (SSPT-BIOAG), Casaccia Research Centre, 00123 Rome, Italy; emilia.pucci@enea.it (E.P.); claudia.zoani@enea.it (C.Z.)

**Keywords:** smart data, search engine concept, search engine visualisation, interoperability, food supply chain, food safety, nutritional quality, traceability, authenticity, food transparency

## Abstract

In recent years, the digital revolution has involved the agrifood sector. However, the use of the most recent technologies is still limited due to poor data management. The integration, organisation and optimised use of smart data provides the basis for intelligent systems, services, solutions and applications for food chain management. With the purpose of integrating data on food quality, safety, traceability, transparency and authenticity, an EOSC-compatible (European Open Science Cloud) traceability search engine concept for data standardisation, interoperability, knowledge extraction, and data reuse, was developed within the framework of the FNS-Cloud project (GA No. 863059). For the developed model, three specific food supply chains were examined (olive oil, milk, and fishery products) in order to collect, integrate, organise and make available data relating to each step of each chain. For every step of each chain, parameters of interest and parameters of influence—related to nutritional quality, food safety, transparency and authenticity—were identified together with their monitoring systems. The developed model can be very useful for all actors involved in the food supply chain, both to have a quick graphical visualisation of the entire supply chain and for searching, finding and re-using available food data and information.

## 1. Introduction

Recently, the food industry has become invested by technological innovations: the route toward the fourth industrial revolution (Industry 4.0) has started [1].

The spread of digitisation and interconnection led to a growth in the quantity of data worldwide and in 1997 the term ‘big data’ was created [2]. There are different definitions of big data in literature because the concept contains several facets. A common understanding is that data is considered as big data if its variety, volume, velocity, veracity, variability, complexity, or value is too hard or impossible to handle with existing data management, data storage or data analysis methods [3,4]. Therefore, new methods are needed to manage, store and analyse big data; this includes new statistical thinking and methods, allowing data itself to identify pertinent variables and patterns that shape the observed outcome [5].

Data are used by companies in several fields: marketing and sales, finance and control, information systems, purchasing, manufacturing, and supply chain [6]. In the agrifood sector, the use of big data is still new [5]. Until now, it has been used to optimise production and to ensure quality and safety [7]; data itself is not useful if it is not used to create value. Today, there are literature debates on how “big” data can become “smart” by transforming the huge quantity of data into strategic knowledge for decision-makers [8].

The rise of the European Open Science Cloud (EOSC) [9] highlights the need for open data sharing, integration, and interoperability, following the FAIR approach [10]. The FAIR principles define that data should be findable, accessible, interoperable, and re-usable. The ambition of EOSC is to develop a web of FAIR data and services for the research community and beyond. The FAIR principles are therefore supporting that data is publicly available for research, industry, consumers, and governments as much as possible. The EOSC turned into an association, and it is collaborating with the European Commission to reach this goal.

The Digital Economy and Society Index (DESI) summarises the digital performances of EU countries in different aspects, such as digital skills, online activity or digital public services, and it tracks changes over time. An analysis of the 2021 index shows a varied picture between member states [11]. This is one reason why digitalisation represents a strategic priority on the political agenda of EU institutions. The use of new technologies gives opportunities to cope with challenges related to the environment, food safety, inclusion, sustainability and transparency of agrifood systems at the national, regional and international level. Digitalisation allows for tracing and tracking across and through the various processes in the chain, enabling greater control over the products and the chain itself, thus promoting consumer confidence in the production system. Information and communication technologies (ICT) support and improve the efficiency of agrifood marketing, product quality and quality maintenance [12].

The main technologies explored in the agrifood sector are smart sensing and monitoring systems (e.g., Internet of Things—IoT, with sensors) using Edge or 5G mobile networks, artificial intelligence (AI), app-based services or blockchain technology (BT) [13,14]. Blockchain was born in 2008 with the cryptocurrency called “Bitcoin” [15], and in the last years it has started to be used by the food industry [16,17]. Its proprieties of decentralisation, transparency and immutability [18] are just some of the strengths that increase research on the topic and its use worldwide [19]. Blockchain was first applied to trace food, but it can also be used, for example, to better monitor food safety, prevent food fraud, reduce waste and give transparency to the consumer [16]. The use of blockchain and big data are proposed to improve agrifood traceability [17,20,21] and to let companies evolve in smart, data driven systems. Transparency and data-sharing between national governments, agencies and industries are the key to better work on risk management, detecting and preventing fraudulent practices and taking actions to inform consumers.

Nowadays, the agrifood sector faces multiple challenges such as population growth, climate change, greenhouse gas emissions, loss of biodiversity, the threat to food security from over-fishing, soil erosion and water shortages. In addition, globalisation in the food trade has led to complexity and fragmentation in the agrifood sector: distances have become bigger and the requirement to keep safety along the supply chain is fundamental to preventing diseases [22]. These challenges currently require a broad vision that should consider all the following interconnected aspects related to food (Figure 1): quality (intended as nutritional and sensorial quality), safety, authenticity, integrity, traceability, transparency and sustainability.

The European Commission aims to assure a high level of food safety and animal and plant health within the EU, through coherent Farm to Fork measures and adequate monitoring, while ensuring an effective internal market with the implementation of a worldwide integrated food safety policy. Of increasing importance is the *One Health* approach, considering the close connection between human health, animal health, and environmental health; therefore food, animal feed, animal and human health, environmental contamination, and environmental impact are closely linked. In order to achieve Goal 2 of the United Nations Sustainable Development Goals (SDGs), another aspect to be considered in every plan and process is sustainability. In the EU, around 88 million tonnes of food waste is generated every year. Primary production, processing and wholesale and retail altogether contribute to the 35% of the total [23].

The sharing of knowledge and information between players allows for the prevention of food losses and surpluses (reducing food waste) and the education of consumers on the conscious use of food.

Safety, quality and authenticity should be guaranteed by sustainability principles. These three concepts should always be taken into consideration to prevent food adulteration and contamination (WHO estimates each year 600 million foodborne illnesses and 420,000 deaths worldwide [24]) to maintain quality during primary production, processing, logistics, retail and post-retail, and to guarantee the nature, identity, claims, and origins of foods. Traceability acquired relevance after a series of food scandals and safety incidents [25]. As a result, today, storing information for tracing in food supply chains has become mandatory worldwide [26], while integrity is a multidisciplinary issue covering all aspects of the food chain from producers to consumers [27].

Existing food nutrition security data, knowledge, and tools for health and agrifood sciences—although widespread—are fragmented, and access is unevenly distributed for users. This means data are not readily FAIR.

This data fragmentation is not only a reality for food security, but also for the whole domain of scientific food data. Nowadays, we are living in a globalised food trade where the competition for innovation and reputation between different academic, governmental, private and new technologies is increasing the amount of data that is being produced. In such a scenario, datasets are widely spread and fragmented, and it is more and more challenging to keep an overview of the existing data that can be re-used for new research and investigations.

This work aims to overcome these challenges by providing a concept for a search engine that will give an overview of the existing datasets covering different food areas and improve food traceability and transparency. The concept was developed in the FNS-Cloud project, and it is not limited to food data but is more generally applicable. Therefore, only the concept is presented while implementation is depending on the data domain. The concept supports the FAIR principles and therefore the idea of open science; it uses some visualisation and it allows for the identification of datasets with information on content, interoperability, quality and accessibility along the food lifecycle. The concept therefore allows for access to and the discovery of data and metadata integration and analysis based on research queries about nutritional quality, food safety, authenticity and transparency to support all actors of the food system: primary producers, processors, distributors, retailers, consumers, researches, inspection and control agencies, certification bodies, authorities and policy makers (Figure 2).

Section 2 describes the methodology and how three food supply chains were investigated to identify the characteristics for a search engine concept. Section 3 presents the resulting search engine concept, and Section 4 draws conclusions about the concept.

## 2. Methodology

The use of data as strategic knowledge in the agrifood sector is a resource for multiple stakeholders. The traceability search engine concept was designed using three specific food supply chains as a model to make sure that the concept is more generally valid considering the differences—sometimes very relevant—that occur among the different production chains. The olive oil, milk and fishery products food chains were selected as a model based on the following criteria:To consider products of both vegetable (olive oil) and animal origin (milk and fishery products);To consider products of interest (e.g., economically important) in various European geographical areas (the Mediterranean area for olive oil, central Europe for milk, northern and Mediterranean area for fishery products);To consider products of different supply chains that allow for covering multiple issues and research questions such as nutrition claims (olive oil), the definition of geographical (milk and fishery products) and botanical (olive oil) origin and production sustainability (fishery products);The possibility to extend the case studies to further products obtained by their processing (e.g., dairy products).

For the olive oil, processes that lead to edible virgin olive oils (extra virgin olive oil and virgin olive oil), as defined in Reg. No 1308/2013 (cons. 2020) [29], were included. For the milk, cow milk was specifically considered without including dairy products. For fishery products, all seawater or freshwater animals (except for live bivalve molluscs, live echinoderms, live tunicates and live marine gastropods, and all mammals, reptiles and frogs) whether wild or farmed were considered including all edible forms, parts and products of such animals, as defined in Reg. No n.853/2004 (cons. 2021) [30]. To cover different types of fishery products three different specific supply chains were examined as examples of three food product categories:Sole supply chain—as representative of a marine wild-caught, medium size, and lean fish;Salmon supply chain—as representative of the aquaculture line, large size, and fatty fish;Anchovies supply chain—as representative of a marine wild-caught, small size, medium fat fish.

First of all, each step of the food chain, with all possible routes, was mapped from primary production to human intake.

The mapping was reported in a flowchart for several purposes: to enable an easier comprehension, to support data analysis step by step, and to support visualisation.

Then, a table was created in which each step was described in detail and the inputs and outputs of the steps were identified. The input and the output of a step describes the status of incoming and outgoing food in order to identify them uniquely.

When applicable, terms were reported with their official definitions (e.g., for FAO, WHO, European Regulations, or EFSA documents).

Finally, every step was examined to understand its effect on nutritional quality, safety, authenticity and transparency, and to obtain a catalogue of significant parameters.

For nutritional quality, intrinsic attributes of food such as chemical composition, physical structure, biochemical changes, nutritional value and nutraceuticals (i.e., the capacity, due to the chemical components, to bring benefit to human health) were considered, as well as shelf-life and the way packaging interacts with the food. For food safety, microbial and chemical contamination were considered (hazards from pathogens, microbial spoilage, presence of mycotoxins, heavy metals, pesticides, etc.). The conditions and practices that influence the nutritional quality of intrinsic attributes of food, and that influence food safety (leading to contamination and foodborne illness) were examined. Authenticity and transparency were considered concerning chemical or genetic markers and profiles that allow for the demonstration of geographical, botanical, or zoologic origin, or for the identification of frauds.

For each criterion (nutritional quality, safety, authenticity and transparency) and each step the following parameters were identified:Parameters of interestParameters of influence

The parameters of interest (data) are considered as analytes (e.g., a chemical, physical or microbiological substance or component) or as nominal properties/characteristics (e.g., profile or taste) that may be subjected to change depending on the conditions in the step under investigation. The parameters of interest were reported for each step, or in some cases for multiple pooled steps, with the specification of the matrix (e.g., semi-finished products). Some of them were compulsorily provided by the manufacturers, while others were measured only voluntarily.

The parameters of influence (metadata) are considered as conditions that can have an effect on or modify the levels of the parameters of interest. They can refer to the matrix of the corresponding stage, or to other aspects (e.g., the environmental, processes, conditions). For example, the physical–chemical characteristics of soil can influence the bioavailability of toxic and potentially toxic elements and therefore their content in the olives; pedoclimatic conditions such as temperature, rainfall and distance from the sea can affect the isotope ratios and nitrogen level and sun exposure can affect the content of polyphenols.

Finally, for each parameter (or class of parameters) and each step, monitoring systems (e.g., indicators/measurement devices) of the parameters of interest and influence were mapped by differentiating between those for offline detection, such as analytic laboratory methodologies, and those permitting in situ and in-line monitoring, such as IoT sensors.

Based on this examination, a concept for a traceability search engine was created that allowed searches for the different aspects described above. Such a concept is presented in the results section.

The above-described methodology has been applied for all of the three supply chains. In order to provide a practical example, what has been elaborated for the olive oil supply chain is presented in Figure 3 (flowchart), Table 1 (inputs and outputs of the supply chain’s steps) and Table 2 (catalogue of parameters of interest and parameters of influence).

## 3. Results: Traceability Search Engine Concept

The purpose of the traceability search engine concept is to support users in searching and finding available food data and information, and to provide knowledge and guidance to its users. A summary of the concept in one sentence could be the following: the traceability search engine concept uses tags having semantical meaning, groups them in dimensions, assigns each data or information resource these tags, allowing use of either a simple search or a visual space search, and offering informed guidance. In the following, some terms will be defined, the idea of dimensions are explained, and finally the visual search of data and information resources is presented.

The first term that the search engine concept uses is data or information resource which can be a dataset or a scientific publication. Such data or information resources can have any form or format, and they can be structured, semi-structured or unstructured data or information. While structured data or information such as databases or Excel files have data organised in a certain order such as rows and columns, unstructured data or information such as free text do not use an ordered structure and data and information must be extracted. Semi-structured data and information is in between structured and unstructured, and it normally combines them. An example could be a Wiki where each term is explained on a separate page and therefore using a list structure, but the content of a page is free text and therefore unstructured. This structural classification is a high-level classification, while more concrete types of data or information resources are commonly used as datasets, databases, Wikis, scientific papers, project reports or websites.

The concept does not require that data and information resources are directly included in a software that implements this concept. It is only required that a resource is described with enough information so that it can be used in the rest of the concept.

The type of a resource is a first characterisation of a data and information resource, and it is considered as a dimension in the concept. A dimension has different tags, and a tag is a term or a keyword that can be assigned to a resource, and it describes a certain aspect of the resource, see Figure 4. The dimension is therefore the group, while the tags are concrete values such as dataset, scientific publication or Wiki.

For each resource, none, one or multiple tags of a dimension can be assigned. The tags and the dimensions have a name and a description explaining their meaning and usage. Additional fields can be added such as input and output as we have seen in the former section. Tags within a dimension can also have a hierarchical structure, allowing for the use of a tree structure with multiple parents. In the food traceability search engine concept, the following dimensions are defined:(a)Type—defines the type of the resource;(b)Food group/matrix—describes a group of foods or a single food item;(c)Food supply chain—describes for each food group/matrix a separate chain of phases;(d)Country—defines the country of origin for the data or information of a resource;(e)Main aspects of food science used in this concept—tags are nutritional quality, safety, and authenticity/traceability (based on the outcome of the former section);(f)Research area—defines a comprehensive list of research aspects in food science which can change over time;(g)Target audience—defines different user groups such as primary producers, processors, distributors, retailers, consumers, researches, inspection and control agencies, certification bodies, authorities and policy makers;(h)Access mode—defines how a resource can be accessed, e.g., open access, restricted access or no access;(i)Year—defines the year of a resource;(j)Chemical substance—defines the list of nutrients, contaminants, and other chemical substances;(k)Parameter of interest—describes properties of foods that are of interest when it comes to traceability in combination with nutritional quality, safety and authenticity/transparency as presented in the former section. Parameters of influence are facts that have an influence on the measurement of the parameter of interest.

The food supply chain is another example of a dimension and each step in the chain is considered a tag. Two or more dimensions can have dependencies between each other. For example, the dimension food group or matrix and the dimension food supply chain. Depending on the food group, the food supply chain can be different as presented in the last section. The concept therefore allows for the defining of dependency rules for tags and dimensions. A dependency rule for dimensions allows for defining each food group’s different food supply chain. Another example is that the parameter of interest is a super dimension where all other dimensions are child dimensions; they were separated because they represent a specific aspect. A dependency rule between tags allows for defining that tags of one dimension can only occur in combination with tags of other dimensions; e.g., certain parameters of interest only occur in combination with a certain phase in the food supply chain, see Table 2.

The description of dimensions and the description of dependency rules are designed to inform users and to serve as a knowledge and documentation base in the domain of the traceability search engine. The documentation should not only allow simple description but also an advanced documentation means as shown in Figure 5. The documentation contains all the information that was collected in the former section, and it represents the current knowledge. As this is developing over time, this documentation needs to be adjusted and extended.

Possible users of such a search engine can range from laypersons to experts, and they are presented in Figure 2. Depending on the type of user, an appropriate scope of information can be made available. For more advanced users, more information can be presented while for less advanced users summaries and simplifications are enough. Therefore, the user should be able to define his/her level of expertise.

The documentation supports consumers by providing information about the production chain of a food item and allowing identification processes that can influence quality, safety, authenticity and transparency. For example, users can check the quality of milk and what affects it. Researchers on the other hand have the ability to find information about which production step influences authenticity and be able to compare their data to other datasets. Food producers also have the possibility to investigate the influences on the quality of their food production, and they can use the knowledge for improvements.

Having assigned the tags to the resources, they can be used to browse and search data and information resources. A simple search allows therefore to select one or multiple tags from one or multiple dimensions and to retrieve all resources that have these tags assigned. The result is normally presented as a list of resources. If more than one tag is used, it should be defined if the AND, OR, or both operators are used. The AND operator defines those resources must have all tags assigned while the OR operator defines that either of the tags must be assigned.

More interesting is the space search because the result list of resources has some limitations. The list, for instance, does not show where no resources are available and comparing tags is more or less comfortable depending on the implementation. A result list must be considered as a keyhole view where only a part is visible, while most of the room is not visible. The space search solves this issue by allowing the use of dimensions to span a result space. If two dimensions are selected, the tags of one dimension are put next to each other on one axis and the tags of the other dimension are put next to each other on the other axis. This results in a table and the data and information resource are listed in the corresponding cell. Table 3 shows a schematic example where the food supply chain was used in combination with three main aspects in food science, which are safety, quality, authenticity and transparency.

The resulting table in Table 3 demonstrates how the keyhole view is removed by showing all possible combinations of two dimensions providing an overview of what data and information resources are available and where no resources are available. The example also shows that not all tags need to be mapped on an axis to reduce the number of columns and rows.

Taking into account different users and their needs, a graphic presentation of the entire supply chain is beneficial, showing its individual steps and the entire food flow process from primary production to human intake. Thanks to this, users can see the entire complexity of the process as well as obtain detailed information about the phase of the process that interests them.

The resulting cells are clickable and, when selected, another view with all resources is presented, showing more information than in the multi-dimensional result space. The idea is that the list items or the result space items represent a short summary, while more information can be found on a separate page when clicking on an item. The list or search space result presentation is called result view while the detail information page is called the detail view. How the detail view is structured and what information it contains depends on the data domain. 

The space search is not limited to two dimensions, but it can combine three or more dimensions. The presentation of the result gets a challenge as multi-dimensional spaces are hard to present and maybe a decomposition in multiple 2-dimensional tables is needed.

The concept also allows for the combining of two or more dimensions on a single axis to increase the space that is spanned and to enlarge the overview of available resources. A limiting factor is the space of the computer screen, in particular if tablets and mobiles are used. In such cases, the reduction of tags mapped on axes is advisable.

Finally, the simple search and the space search can be combined. While the space search presents the results in multi-dimensional space, the dimensions that were not used to span the space can be used to further filter resources. In this way, more advanced search operations are possible and more specific results can be presented.

## 4. Conclusions

The traceability search engine concept and its developed model fit the purpose of collecting, organising, making available and integrating data and metadata on food quality, safety, traceability, transparency, and the authenticity of products along the food supply chain, following the FAIR approach. The developed model is helpful both to have a graphical visualisation of the entire food supply chain and to have the possibility to carry out different types of searches. Searches can be made on different dimensions alone or in combination between them (type of resource, food/matrix group, food chain, country, aspects related to food science, research area, target audience, access mode, year, chemistry, the parameter of interest) and their different tags (e.g., a step in a specific food supply chain). This model supports users in finding available food data and information, and it provides them with knowledge and guidance, according to the type of user. Indeed, depending on the expertise of the user, much more detailed information can be made available for advanced users and simplifications or summaries can be delivered for less advanced users. Sharing smart data in the network can support all actors in the food system. Thanks to the dedicated information displayed for each user category, companies, policy makers, local authorities and citizens can benefit from the model. This work integrates knowledge of food science and innovative engineering. The next step will be exploring the possibility to integrate blockchain technologies in the demonstrator to give more transparency to all users.

## Figures and Tables

**Figure 1 foods-11-00989-f001:**
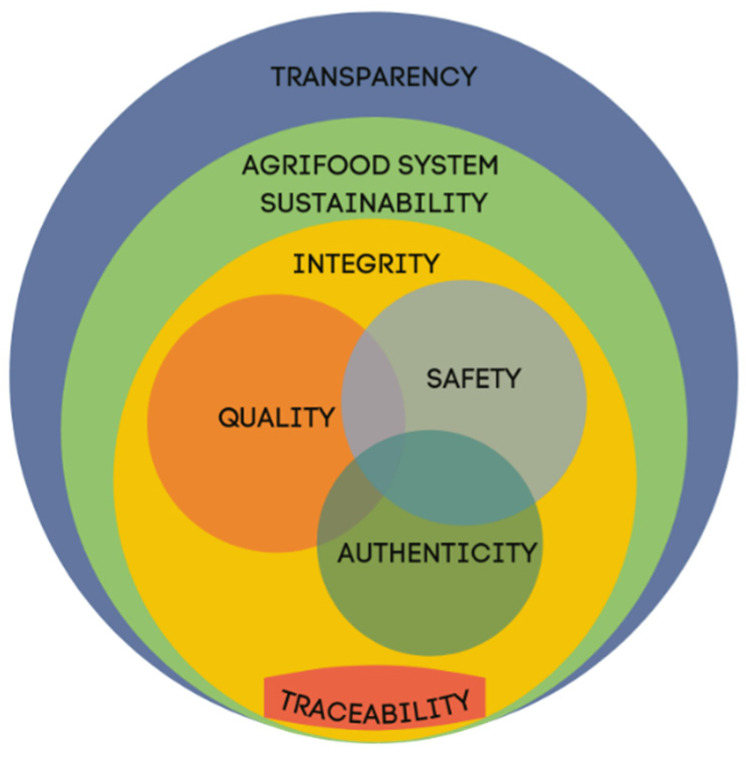
Relationship between food transparency, food integrity, food traceability, food quality, food safety, food authenticity and sustainability.

**Figure 2 foods-11-00989-f002:**
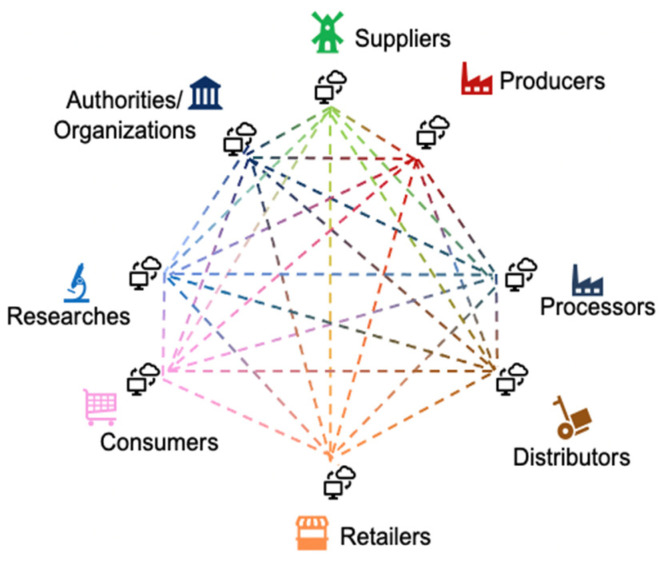
Interoperability network through food system [28].

**Figure 3 foods-11-00989-f003:**
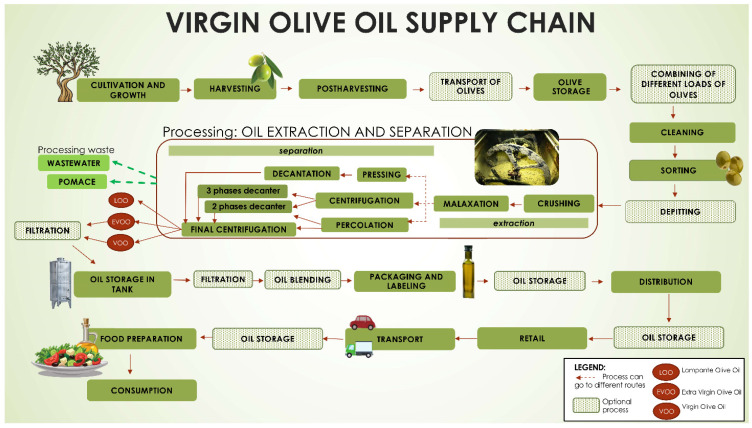
Flowchart of the edible virgin olive oils supply chain.

**Figure 4 foods-11-00989-f004:**
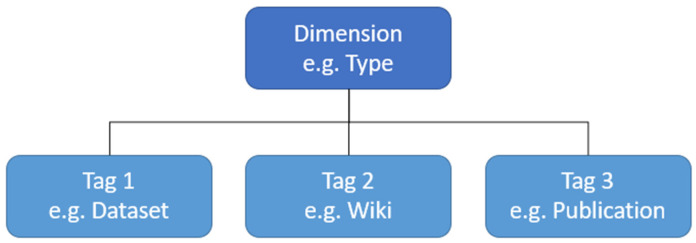
A dimension is a group of tags which are possible concrete terms or keywords of the dimension.

**Figure 5 foods-11-00989-f005:**
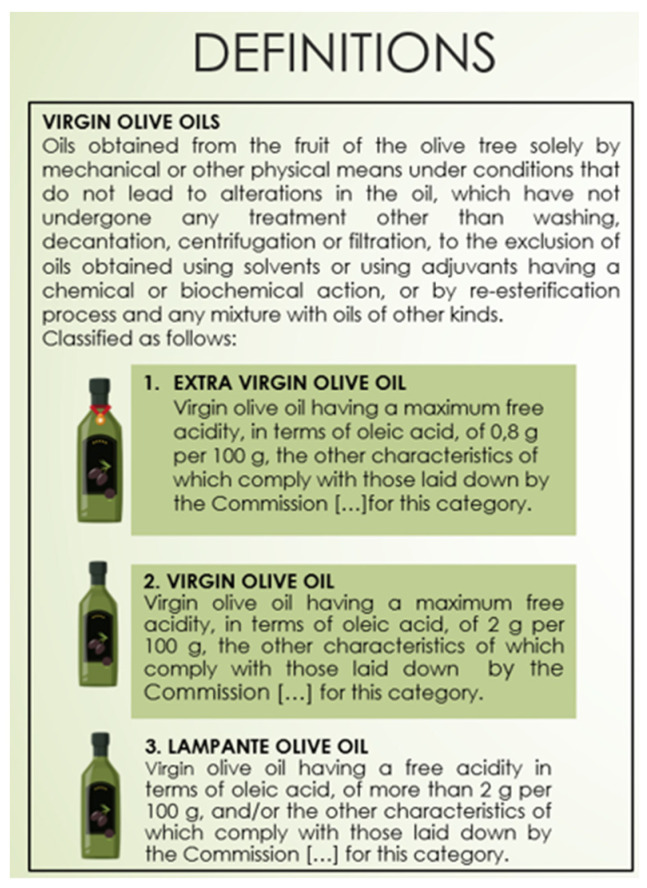
Example of detailed documentation of the olive oil milling process.

**Table 1 foods-11-00989-t001:** Extract of the table describing each step with their inputs and outputs.

Step	Definition	Input	Output
Cultivation and Growth	All stages that concern agronomic practices to make olives grow and keep them healthy until harvest	Olive Trees	Olives
Harvesting	The process of gathering a ripe crop from olives fields. It can be done after natural fall, by hand, by beating the branches, with shakers, by combing (previously is commonly used to put canvases on the soil for the reception of the harvested fruits)	Olives	Olives
Postharvesting	Olives are taken from the nets on the ground and put into bins	Olives	Olives
Transport of Olives	Olives are transported to oil mill by olive grower	Olives	Olives
Olive Storage	Olives are stored in rigid and ventilated containers in a cool and dry environment	Olives	Olives
Combining Different Loads of Olives	Olives can arrive from different olive’s growers and are mixed together	Olives	Olives
Cleaning	Involves defoliation and washing	Olives	Olives
Sorting	Discarding any bruised or defective fruit	Olives	Olives
Depitting	Separation of the pits from the olives	Olives	Olives

**Table 2 foods-11-00989-t002:** Extract of the catalogue of parameters of interest and parameters of influence for nutritional quality, safety and authenticity/transparency concerning the virgin olive oil supply chain. The phases in italics are optional and at the choice of the food companies and “X” indicate that the process phase does not affect authenticity/traceability.

		Nutritional Quality		Safety		Authenticity/Transparency	
Phase	Matrix	Parameters of Interest	Parameters of Influence	Parameters of Interest	Parameters of Influence	Parameters of Interest	Parameters of Influence
Cultivation and Growth	Olives	fatty acids (FFAs, SFAs, MUFAs and PUFAs), total polyphenols, tocopherol, secoiridoids (oleuropein, hydroxytyrosol), phytosterols, pigments (carotenoids, chlorophylls), lignans, secoiridoid derivatives, 3,4-DHPEA-AC, monoglycerides and peroxides, DAGs, peroxide value, pH, total CHO, soluble solids, % in oil	climatic and pedoclimatic conditions (e.g., air composition, sun exposure, physical-chemical characteristics of soil and trees, irrigation); type and fertilisers content; pruning, pest and disease management	toxic and potentially toxic elements, Polycyclic Aromatic Hydrocarbons (PAHs), mycotoxins, radionuclides	pedoclimatic conditions (e.g., physical-chemical characteristics of soil, environmental pollution) physiopatological factors, biocides and plant protection products (pesticides used)	isotopic ratios, rare earth elements, micronutrients, pigments profiles, genomic profiles	cultivar, latitude, longitude, rainfall, distance from sea, sun exposition, physical-chemical characteristics of soil, fertilisers use
Harvesting			time (t), techniques applied, maturity index, detachment index	foreign matters, texture, integrity	t, harvesting system, microbiological and biological contaminants	X	X
Postharvesting			Temperature (T), t, mechanical breakages, equipment	toxic and potentially toxic elements, free acidity, peroxide, K232 value, mold, insect and microbial infection, mold, FFA, peroxide value	aeration, equipment and storage conditions (T, t), mechanical breakages, handling, foreign materials		
*Transport of Olives*							
Olives Storage		micronutrients content, free acidity level, peroxide, K232 value, K270 value, mold	storage conditions (T, t), processing equipment (e.g., tanks, pipes, drums, etc.)				
Arrival at the Mill		air humidity, free acidity	storage conditions (T, t)				
*Combining Different Loads of Olives*		micronutrients, total polyphenols, secoiridoids (oleuropein, hydroxytyrosol) phytosterols, pigments (e.g., carotenoids)	mixing ratio, content in each single load of olives	chemical residues	mixing ratio, content in each single load of olives	isotopic ratios, rare earth elements, micronutrients, pigments profiles, genomic profiles	olives loads provenance, cultivar, latitude, longitude, rainfall, distance from sea, sun exposition, physical-chemical characteristics of soil, fertilisers use
Cleaning		total polyphenols	T, t, washing water quality	toxic and potentially toxic elements, foreign matters, pesticides	T, t, washing water quality	X	X
Sorting		olive texture	X	olives texture	X		
*Depitting*		pits, pit dust	machines efficiency	pit dust	pit hardness		

**Table 3 foods-11-00989-t003:** Example of result presentation of a 2-dimensional space search.

	Primary Production	Processing	Packaging	Storage and Distribution	Retail	Final Consumption
**Safety**						TDS data (link) Medication concentration data (link)
**Nutritional** **quality**		Possible contamination during olive oil extraction (link)		Loss of vitamins during storage (link)	Label information (link)	Greece food composition database (link)
**Authenticity/transparency**	Isotope data (link)					

## Data Availability

Data is contained within the article.

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
