# Peer review of "A Search Engine Concept to Improve Food Traceability and Transparency: Preliminary Results"

_foods, 2022, doi:10.3390/foods11070989_

Round 1
Reviewer 1 Report
Interesting work. I made some comments in the attached file.

Author Response
Dear Editor,
On behalf of the other authors and myself, I would like to sincerely thank the Reviewers for their valuable comments, which encouraged us about the overall merits of our manuscript and highlighted its weaknesses, helping us to improve our work. All the issues and the concerns raised by the Reviewers have been taken into due consideration and carefully addressed.
In the attached file you can find listed all the comments of the reviewers along with our answers and the indication of the corresponding changes in the revised texts. All revisions were entered using the "Track Changes” function, as requested by the editors.
We hope that, in this revised version, the manuscript is suitable for publication in the Special Issue of Foods.
Kind regards,
Caterina Palocci
Response to Reviewer 1 Comments
Point 1: Please indicate that cases and deaths are of food illnesses
Response 1: We sincerely thank the Reviewer for highlighting this error, obviously these cases and deaths are due to food illnesses. Text was amended as follows: WHO estimates each year 600 million foodborne illnesses and 420,000 deaths worldwide.
Point 2: produced. In...
Response 2: We sincerely thank the Reviewer for highlighting this typesetting error. Text was amended according to Reviewer’s suggestion.
Point 3: Since the search engine was designed focused on 3 specific food supply chais used as model it seems odd that dairy and fishery products are not discussed in the results section. I believe that this was decided because it was considered that just one example of extracted information was enough to exemplify the performance of the search engine. However, it seems strange not to have any mention to the other supply chains.
Response 3: The Reviewer is definitely right. The traceability search engine concept was designed using three specific food supply chains as a model (olive oil, milk, and fishery products) to make sure that the concept is more generally valid considering the differences - sometimes very relevant - that occur among the different pro-duction chains. The same methodology has been applied for all three supply chains. The olive oil production chain is shown as a practical example of what has been done and the results obtained.
Point 4: consider using italic
Response 4: We sincerely thank the Reviewer for highlighting this typesetting error. Text was amended according to Reviewer’s suggestion.
Point 5: please refer in legen that operations in italica are optional
Response 5: The Reviewer is definitely right. The legend has been changed according to Reviewer’s suggestion. Text was amended as follows: The phases in italics are optional and at the choice of the food companies.
Point 6: E.g.
Response 6: We sincerely thank the Reviewer for highlighting this typesetting error. Text was amended according to Reviewer’s suggestion.
Point 7: the number of...
Response 7: We sincerely thank the Reviewer for highlighting this typesetting error. Text was amended according to Reviewer’s suggestion.
Point 8: please clarify sentence: Independent of the simple or space search is used, resulting resources can also be selected and all available information of the resources are displayed to the user.
Response 8: We sincerely thank the Reviewer for this comment. Sentence was rephrased to make it more clear as follows: “The idea is that the list items or the result space items represent a short summary while more information can be found on a separate page when clicking on an item. The list or search space result presentation is called result view while the detail information page is called the detail view. How the detail view is structured and what information it contains, is depending on the data domain”.
Point 9: two dimensions
Response 9: We sincerely thank the Reviewer for highlighting this typesetting error. Text was amended according to Reviewer’s suggestion.
Reviewer 2 Report
The paper seems a preliminary result of the above-mentioned European FNS-Cloud project: if so it should be highlighted, if not it needs to be well argued. In any case, the objective of the paper is of particular interest to the scientific community but the paper needs some important improvement.
Major Revision
First of all it is not clear the structure of the software concept (at least for those readers less experienced in this subject) or somehow confusing. In fact the Methodology section, even well describing (with few imperfections: see minor revision below) the chosen model (olive oil chain) does not provide any information about search and data integration techniques used (or assumed to be used) in this concept model. For example, in Figure 4 (see below: tables should referenced as tables and not figures) the only inputs and outputs are olives (no input is present in the first step!): it is not clear why authors limit so strongly input and output presentation (even for the less experienced user this appears unusual). This is true for the so called "Parameters of Interest", but even more for the "Parameters of Influence": some information should be provided (few examples are more than enough) about how some parameters could influence other features, even in future steps.
With regards to the Results section, its first part seems a continued and more detailed continuation of the Methodology section. On the contrary, authors should provide more examples as that shown in Figure 7 (moreover in this particular case authors should give an exact description of the assumed user as only limited information is shown as search result).
Minor Revision:
Figures 4, 5 and 7 are tables and not figures
Discussion section should be renamed Conclusions
Author Response
Dear Editor,
On behalf of the other authors and myself, I would like to sincerely thank the Reviewers for their valuable comments, which encouraged us about the overall merits of our manuscript and highlighted its weaknesses, helping us to improve our work. All the issues and the concerns raised by the Reviewers have been taken into due consideration and carefully addressed.
In the attached file you can find listed all the comments of the reviewers along with our answers and the indication of the corresponding changes in the revised texts. All revisions were entered using the "Track Changes” function, as requested by the editors.
We hope that, in this revised version, the manuscript is suitable for publication in the Special Issue of Foods.
Kind regards,
Caterina Palocci
Response to Reviewer 2 Comments
Point 1: The paper seems a preliminary result of the above-mentioned European FNS-Cloud project: if so it should be highlighted, if not it needs to be well argued. In any case, the objective of the paper is of particular interest to the scientific community but the paper needs some important improvement.
Response 1: We sincerely thank the Reviewer for his/her words of appreciation regarding the objective of the paper. The relation to FNS-Cloud was added and it is justified why only the concept is presented.
Point 2: First of all it is not clear the structure of the software concept (at least for those readers less experienced in this subject) or somehow confusing. In fact the Methodology section, even well describing (with few imperfections: see minor revision below) the chosen model (olive oil chain) does not provide any information about search and data integration techniques used (or assumed to be used) in this concept model. For example, in Figure 4 (see below: tables should referenced as tables and not figures) the only inputs and outputs are olives (no input is present in the first step!): it is not clear why authors limit so strongly input and output presentation (even for the less experienced user this appears unusual). This is true for the so called "Parameters of Interest", but even more for the "Parameters of Influence": some information should be provided (few examples are more than enough) about how some parameters could influence other features, even in future steps.
Response 2: We understand the Reviewer’s perplexities. Two sentences were added at the end of section 1 to provide an overiew about section 2 and 3. Additional description about the structure of the concept were added to section 3. In addition, section 2 has been amended to clarify the description of the methodology applied to all supply chains considered. In table 1 (Figure 4) the input has been inserted in the first step. The descriprion of "Parameters of Influence" has been implemented with the following examples: “For example, physical-chemical characteristics of soil can influence the bioavailability of toxic and potentially toxic elements and therefore their content in the olives; pedoclimatic conditions such as temperature, rainfall and distance from sea can affect the isotope ratios and nitrogen level and sun exposure can affect the content of polyphenols”.
Point 3: With regards to the Results section, its first part seems a continued and more detailed continuation of the Methodology section. On the contrary, authors should provide more examples as that shown in Figure 7 (moreover in this particular case authors should give an exact description of the assumed user as only limited information is shown as search result).
Response 3: We understand the Reviewer’s perplexities. Description was added at the beginning of section 3 to descript the structure of the search engine concept. One more figure was added to explain dimensions and tags. Users and possible information that can be shown on result view and detail view are described in more detail.
Point 4: Figures 4, 5 and 7 are tables and not figures
Response 4: The Reviewer is definitely right. Figures 4, 5 and 7 are tables, then renamed Table 1, Table 2 and Table 3 respectively,.
Point 5: Discussion section should be renamed Conclusions
Response 5: According to Reviewer’s suggestion the discussion section was renamed conclusions.